# Exemplar Guided Active Learning

**Jason Hartford**
AI21 Labs
jasonh@cs.ubc.ca

**Kevin Leyton-Brown**
AI21 Labs
kevinlb@cs.ubc.ca

**Hadas Raviv**
AI21 Labs
hadasr@ai21.com

**Dan Padnos**
AI21 Labs
danp@ai21.com

**Shahar Lev**
AI21 Labs
shaharl@ai21.com

**Barak Lenz**
AI21 Labs
barakl@ai21.com

## Abstract

We consider the problem of wisely using a limited budget to label a small subset of a large unlabeled dataset. We are motivated by the NLP problem of word sense disambiguation. For any word, we have a set of candidate labels from a knowledge base, but the label set is not necessarily representative of what occurs in the data: there may exist labels in the knowledge base that very rarely occur in the corpus because the sense is rare in modern English; and conversely there may exist true labels that do not exist in our knowledge base. Our aim is to obtain a classifier that performs as well as possible on examples of each "common class" that occurs with frequency above a given threshold in the unlabeled set while annotating as few examples as possible from "rare classes" whose labels occur with less than this frequency. The challenge is that we are not informed which labels are common and which are rare, and the true label distribution may exhibit extreme skew. We describe an active learning approach that (1) explicitly searches for rare classes by leveraging the contextual embedding spaces provided by modern language models, and (2) incorporates a stopping rule that ignores classes once we prove that they occur below our target threshold with high probability. We prove that our algorithm only costs logarithmically more than a hypothetical approach that knows all true label frequencies and show experimentally that incorporating automated search can significantly reduce the number of samples needed to reach target accuracy levels.

## 1 Introduction

We are motivated by the problem of labelling a dataset for word sense disambiguation, where we want to use a limited budget to collect annotations for a reasonable number of examples of each sense for each word. This task can be thought of as an active learning problem (Settles, 2012), but with two nonstandard challenges. First, for any given word we can get a set of candidate labels from a knowledge base such as WordNet (Fellbaum, 1998). However, this label set is not necessarily representative of what occurs in the data: there may exist labels in the knowledge base that do not occur in the corpus because the sense is rare in modern English; conversely, there may also exist true labels that do not exist in our knowledge base. For example, consider the word "bass." It is frequently used as a noun or modifier, e.g., "the *bass* and alto are good singers", or "I play the *bass* guitar". It is also commonly used to refer to a type of fish, but because music is so widely discussed online, the fish sense of the word is orders of magnitude less common than the low-frequency sound sense in internet text. The Oxford dictionary (Lexico) also notes that bass once referred to a fibrous material used in matting or chords, but that sense is not common in modern English. We want a method that collects balanced labels for the common senses, "*bass* frequencies" and "*bass* fish", and ignores sufficiently rare senses, such as "fibrous material". Second, the empirical distribution of the

true labels may exhibit extreme skew: word sense usage is often power-law distributed (McCarthy et al., 2007) with frequent senses occurring orders of magnitudes more often than rare senses.

When considered individually, neither of these constraints is incompatible with existing active learning approaches: incomplete label sets do not pose a problem for any method that relies on classifier uncertainty for exploration (new classes are simply added to the classifier as they are discovered); and extreme skew in label distributions has been studied under the guided learning framework wherein annotators are asked to explicitly search for examples of rare classes rather than simply label examples presented by the system (Attenberg & Provost, 2010). But taken together, these constraints make standard approaches impractical. Search-based ideas from guided learning are far more sample efficient with a skewed label distribution, but they require both a mechanism through which annotators can search for examples and a correct label set because it is undesirable to ask annotators to find examples that do not actually occur in a corpus.

Our approach is as follows. We introduce a frequency threshold, $\gamma$, below which a sense will be deemed to be "sufficiently rare" to be ignored (i.e. for sense $y$, if $P(Y = y) = p_y < \gamma$ the sense is rare); otherwise it is a "common" sense of the word for which we want a balanced labeling with other common senses. Of course, we do not know $p_y$, so it must be estimated online. We do this by providing a stopping rule that stops searching for a given sense when we can show with high probability that it is sufficiently rare in the corpus. We automate the search for rare senses by leveraging the high-quality feature spaces provided by modern self-supervised learning approaches (Devlin et al., 2018; Radford et al., 2019; Raffel et al., 2019). We leverage the fact that one typically has access to a single example usage of each word sense[1], which enables us to search for more examples of a sense in a local neighborhood of the embedding space. This allows us to develop a hybrid guided and active learning approach that automates the guided learning search procedure. Automating the search procedure makes the method cheaper (because annotators do not have to explicitly search) and allows us to maintain an estimate of $\hat{p}_y$ by using importance-weighted samples. Once we have found examples of common classes, we switch to more standard active learning methods to find additional examples to reduce classifier uncertainty.

Overall, this paper makes two key contributions. First, we present an Exemplar Guided Active Learning (EGAL) algorithm that offers strong empirical performance under extremely skewed label distributions by leveraging exemplar embeddings. Second, we identify a stopping rule that makes EGAL robust to misspecified label sets and prove that this robustness only imposes a logarithmic cost over a hypothetical approach that knows the correct label set. Beyond these key contributions, we also present a new Reddit word sense disambiguation dataset, which is designed to evaluate active learning methods for highly skewed label distributions.

## 2  Related Work

**Active learning under class imbalance**  The decision-boundary-seeking behavior of standard active learning methods which are driven by classifier uncertainty has a class balancing effect under moderately skew data (Attenberg & Ertekin, 2013). But, under extreme class imbalance, these methods may exhaust their labeling budgets before they ever encounter a single example of the rare classes. This issue is caused by an epistemic problem: the methods are driven by classifier uncertainty, but standard classifiers cannot be uncertain about classes that they have never observed. Guided learning methods (Attenberg & Provost, 2010) address this by assuming that annotators can explicitly search for rare classes using a search engine (or some other external mechanism). Search may be more expensive than annotation, but the tradeoff is worthwhile under sufficient class imbalance. However, explicit search is not realistic in our setting: search engines do not provide a mechanism for searching for a particular sense of a word and we care about recovering all classes that occur in our dataset with frequency above $\gamma$, so searching by sampling uniformly at random would require labelling $n \geq \mathcal{O}(\frac{1}{\gamma^2})$ samples[2] to find all such classes with high probability.

Active learning with extreme class imbalance has also been studied under the "active search" paradigm (Jiang et al., 2019) that seeks to find as many examples of the rare class as possible in a finite budget

of time, rather than minimizing classifier uncertainty. Our approach instead separates explicit search from uncertainty minimization in two different phases of the algorithm.

**Active learning for word sense disambiguation**  Many authors have showed that active learning is a useful tool for collecting annotated examples for the word sense disambiguation task. Chen et al. (2006) showed that entropy and margin-based methods offer significant improvements over random sampling. To our knowledge, Zhu & Hovy (2007) were the first to discuss the practical aspects of highly skewed sense distributions and their effect on the active learning problems. They studied over- and under-sampling techniques which are useful once one has examples, but did not address the problem of finding initial points under extremely skewed distributions.

Zhu et al. (2008) and Dligach & Palmer (2011) respectively share the two key observations of our paper: good initializations lead to good active learning performance and language models are useful for providing a good initialization. Our work modernizes these earlier papers by leveraging recent advances in self-supervised learning. The strong generalization provided by large-scale pre-trained embeddings allow us to guide the initial search for rare classes with exemplar sentences which are not drawn from the training set. We also provide stopping rules that allow our approach to be run without the need to carefully select the target label set, which makes it practical run in an automated fashion.

Yuan et al. (2016) also leverage embeddings but they use label propagation to nearest neighbors in embedding space. This approach is similar to ours in that it also uses self-supervised learning, but we have access to ground truth through the labelling oracle which offers some protection against the possibility that senses are poorly clustered in embedding space.

**Pre-trained representations for downstream NLP tasks**  There are a large number of recent papers showing that combining extremely large datasets with large Transformer models (Vaswani et al., 2017) and training them on simple sequence prediction objectives leads to contextual embeddings that are very useful for a variety of downstream tasks. In this paper we use contextual embeddings from BERT (Devlin et al., 2018) but because the only property we leverage is the fact that the contextual embeddings provide a useful notion of distance between word senses, the techniques described are compatible with any of the recent contextual models (e.g. Radford et al., 2019; Raffel et al., 2019).

## 3   Exemplar-guided active learning

We are given a large training set of unlabeled examples described by features (typically provided by an embedding function), $X_i \in \mathbb{R}^d$, and our task is to build a classifier, $f : \mathbb{R}^d \to \{1, \ldots, K\}$, that maps a given example to one of $K$ classes. We are evaluated based on the accuracy of our trained classifiers on a *balanced* test set of the "common classes": those classes, $y_i$, that occur in our corpus with frequency, $p_{y_i} > \gamma$, where $\gamma$ is a known threshold. Given access to some labelling oracle, $l : \mathbb{R}^d \to \{1, \ldots, K\}$, that can supply the true label of any given example at a fixed cost, we aim to spend our labelling budget on a set of training examples such that our resulting classifier minimizes the $0 - 1$ loss on the $k(\gamma) = \sum_i \mathbf{1}\big[p_{y_i} \geq \gamma\big]$ classes that exceed the threshold, $\mathcal{L} = \frac{1}{k(\gamma)} \sum_{i=1}^{K} \mathbf{1}\big[p_{y_i} \geq \gamma\big] \mathbf{E}_{X:l(X)=y_k}[\mathbf{1}(f(X) \neq y_k)]$.

That is, any label that occurs with probability at least $\gamma$ in the observed data generating process will receive equal weight, $\frac{1}{k(\gamma)}$, in the test set and anything that occurs less frequently can be ignored. The task is challenging because rare classes (i.e. those which occur with frequency $\gamma < p_y \ll 1$) are unlikely to be found by random sampling, but still contribute a $\frac{1}{k(\gamma)}$ fraction of the overall loss.

Our approach leverages the guided learning insight that "search beats labelling under extreme skew", but automates the search procedure. We assume we have access to high-quality embeddings—such as those from a modern statistical language model—which gives us a way of measuring distances between word usage in different contexts. If these distances capture word senses and we have an example usage of the word for each sense, a natural search strategy is to examine the usage of the word in a neighborhood of the example sentence. This is the key idea behind our approach: we have a search phase where we sample the neighborhood of the exemplar embedding until we find at least one example of each word sense, followed by a more standard active learning phase where we seek examples that reduce classifier uncertainty.

**Input :** $\mathcal{D} = \{X_i\}_{i \in 1\ldots n}$ a dataset of unlabeled examples
        $\phi : \text{domain}(X) \to \mathbb{R}^d$, $d : \mathbb{R}^d \times \mathbb{R}^d \to \mathbb{R}$ an embedding and distance function
        $l : X_i \to y_i$ a labeling operation (such as querying a human expert)
        $L$ : The total number of potential class labels
        $\gamma$ : the label-probability threshold
        $E$ the set of exemplars, $|E| = L$
        $B$ : a budget for maximum number of queries
        $b$ : batch size of queries sampled before the model is retrained

$\mathcal{A} \leftarrow \{1, \ldots, L\}$ # Set of active classes
$\mathcal{C} \leftarrow \emptyset$ # Set of completed classes
$\mathcal{D}^{(l)} \leftarrow \emptyset$ # Set of labeled examples
**while** $|\mathcal{D}^{(l)}| < B$ **do**
    $\mathcal{A}' = \mathcal{A}$ # $\mathcal{A}'$ is the target set of classes for the algorithm to find.
    **while** $\mathcal{A}' \neq \emptyset$ *and number of collected samples* $< b$ **do**
        Select random $i'$ from $\mathcal{A}'$ and set $\mathcal{A}' \leftarrow \mathcal{A}' \setminus \{i'\}$ and $X \leftarrow \emptyset$
        **repeat**
            $X \leftarrow X \cup \{x\}$ where $x$ is selected with exemplar $E_{i'}$ using either equation 1 or
               $\epsilon$-greedy sampling.
            $y \leftarrow \{l(x) \text{ for x in } X\}$ # Label each example in $X$
        **until** *(Number of unique labels in $y = \lfloor b/L \rfloor$) or (Number of labeled samples = b)*;
        Update empirical means $\hat{p}_y$ and remove any classes with $\hat{p}_y + \sigma_y < \gamma$ from $\mathcal{A}$ and $\mathcal{A}'$
        $\mathcal{D}^{(l)} \leftarrow \mathcal{D}^{(l)} \cup (X, y)$
        $\mathcal{A} \leftarrow \{i \in \mathcal{A} : i \text{ not in unique labels in } D^{(l)}\}$ # Remove observed labels from the active
          set
    **end**
    Sample the remainder of the batch, $(X, y)$, using using either algorithm in equation 3
    $\mathcal{D}^{(l)} \leftarrow \mathcal{D}^{(l)} \cup (X, y)$
    Update empirical means $\hat{p}_y$ and remove any classes with $\hat{p}_y + \sigma_y < \gamma$ from $\mathcal{A}$
    $\mathcal{A} \leftarrow \{i \in \mathcal{A} : i \text{ not in unique labels in } D^{(l)}\}$
    Update classifier using $\mathcal{D}^{(l)}$.
**end**

**Algorithm 1:** EGAL: Exemplar Guided Active Learning

In the description below we denote the embedding vector associated with the target word in sentence, $i$, as $x_i$. For each sense, $y$, denote the embedding of an example usage as $\tilde{x}_y$. We assume that this example usage is selected from a dictionary or knowledge base so we do not include it in our classifier's training set. Full pseudocode is given in Algorithm 1.

**Guided search** Given an example embedding, $\tilde{x}_y$, we could search for similar usages of the word in our corpus by iterating over corpus examples, $x_i$, sorted by distance, $d_i = \|x_i - \tilde{x}_y\|_2$. However, using this approach does not give us a way of maintaining an estimate of $\hat{p}_y$, the empirical frequency of the word sense in the corpus which we need for our stopping rule that stops searching for classes that are unlikely to exist in the corpus. Instead we sample each example to label, $x_i$, from a Boltzmann distribution over unlabeled examples,

$$x_i : i \sim \text{Cat}(q = [q_1, \ldots, q_n]), \ q_i = \frac{\exp(-d_i/\lambda_y)}{\sum_i \exp(-d_i/\lambda_y)}, \tag{1}$$

where $\lambda_y$ is a temperature hyper-parameter that controls the sharpness of $q$.

We sample with replacement and maintain a count vector $c$ that tracks the number of times an example has been sampled. If an example has previously been sampled, labelling does not diminish our annotation budget because we can simply look up the label, but maintaining these counts lets us maintain an unbiased estimate of $p_y$, the empirical frequency of the sense label and gives a way of choosing $\lambda_y$, the length scale hyper-parameter, which we describe below. We continue drawing samples until we have a batch of $b$ labeled examples.

**Optimizing the length scale**  The sampling procedure selects examples to label in proportion to how similar they are to our example sentence, where similarity is measured by a square exponential kernel on the distance $d_i$. To use this, we need to choose a length scale, $\lambda_y$, which selects how to scale distances such that most of the weight is applied to examples that are close in embedding space. One challenge is that embedding spaces may vary across words and in density around different example sentences. If $\lambda_y$ is set either too large or too small, one tends to sample few examples from the target class because for extreme values of $\lambda$, $q$ either concentrates on a single example (and is uniform over the rest) or is uniform over all examples. We address this with a heuristic that automatically selects the length scale for each example sentence $x_y$. We choose $\lambda$ that minimizes

$$\lambda_y = \arg\min_\lambda \mathbb{E}\left[\frac{1}{\sum_{i \in B} w_i^2}\right]; \ \ w_i = \frac{c_i(x_y)}{\sum_{j \in B} c_j(x_y)}. \tag{2}$$

This score measures the effective sample size that results from sampling a batch of $B$ examples for example sentence $x_y$. The score is minimized when $\lambda$ is set such that as much probability mass as possible is placed on a small cluster of examples. This gives the desired sampling behavior of searching a tight neighborhood of the exemplar embedding. Because the expectation can be estimated using only unlabeled examples, we can optimize this by averaging the counts from multiple runs of the sampling procedure and finding the minimal score by binary search.

**Active learning**  The second phase of our algorithm builds on standard active learning methods. Most active learning algorithms select unlabeled points to label, ranking them by an "informativeness" score for various notions of informativeness. The most widely used scores are the *uncertainty sampling* approaches, such as *entropy* sampling, which scores examples by $s_{\text{ENT}}$, the entropy of the classifier predictions, and the *least confidence* heuristic $s_{\text{LC}}$, which selects the unlabeled example about which the classifier is least confident. They are defined as

$$s_{\text{ENT}}(x) = -\sum_i P(y_i|x;\theta) \log P(y_i|x;\theta); \quad s_{\text{LC}}(x) = -\max_y P(y|x;\theta). \tag{3}$$

Typically examples are selected to maximize these scores, $x_i = \arg\max_{x \in \mathcal{X}_{\text{pool}}} s(x)$, but again we can sample from a distribution implied by the score function to select examples $x_i : i \sim \text{Cat}(q' = [q'_1, \ldots, q'_n])$ and maintain an estimate of $p_y$ in a manner analogous to Equation 2 as $q'_i = \frac{\exp(s_{LC}(x_i)/\lambda_y)}{\sum_i \exp(s_{LC}(x_i)/\lambda_y)}$,

**$\epsilon$-greedy sampling**  The length scale parameters for Boltzmann sampling can be tricky to tune when applied to the active learning scores because the scores vary over the duration of training. The means that we cannot use the optimization heuristic that we applied to the guided search distribution. A simple alternative to the Boltzmann distribution sampling procedure is to sample some $u \sim \text{Uniform}(0,1)$ and select a random example whenever $u \leq \epsilon$. $\epsilon$-greedy sampling is far simpler to tune and analyze theoretically, but has the disadvantage that one can only use the random steps to update estimates of the class frequencies. We evaluate both methods in the experiments.

**Stopping conditions**  For every sample we draw, we estimate the empirical frequencies of the senses. We continue to search for examples of each sense as long as an upper bound on the sense frequency exceeds our threshold $\gamma$. For each sense, the algorithm remains in "importance weighted search mode" as long as $\hat{p}_y + \sigma_y > \gamma$ and we have not yet found an example of sense $y$ in the unlabeled pool. Once we have found an example of $y$, we stop searching for more examples and instead let further exploration be driven by classifier uncertainty.

Because any wasted exploration is costly in the active learning setting, a key consideration for the stopping rule is choosing the confidence bound to be as tight as possible while still maintaining the required probabilistic guarantees. We use two different confidence bounds for each of the sampling strategies. When sampling using the $\epsilon$-greedy strategy, we know that the random variable, $y$, obtains values in $\{0, 1\}$ so we can get tight bounds on $p_y$ using Chernoff's bound on Bernoulli random variables. We use the following implication of the Chernoff bound (see Lattimore & Szepesvári, 2020, chapter 10),

**Lemma 1** (Chernoff bound). *Let $y_i$ be a sequence of Bernoulli random variables with parameter $p_y$, $\hat{p}_y = \frac{1}{n}\sum_{i=1}^n y_i$ and $KL(p, q) = p\log(p/q) + (1-p)\log((1-p)/(1-q))$. For any $\delta \in (0, 1)$ we*

*can define the upper and lower bounds as follows,*

$$U(\delta) = \max\{x \in [0,1] : KL(\hat{p}_y, x) \leq \frac{\log(1/\delta)}{n}\}, \; L(\delta) = \min\{x \in [0,1] : KL(\hat{p}_y, x) \leq \frac{\log(1/\delta)}{n}\}$$

*and we have that* $\mathbb{P}\left[p_y \geq U(\delta)\right] \leq \delta$ *and* $\mathbb{P}\left[p_y \leq L(\delta)\right] \leq \delta$.

There do not exist closed-form expressions for these upper and lower bounds, but they are simple bounded 1D convex optimization procedures that can be solved efficiently using a variety of optimization methods. In practice we use Scipy's (Virtanen et al., 2020) implementation of Brent's method (Brent, 1973).

When using the importance weighted approach, we have to contend with the fact that the random variables implied by the importance-weighted samples are not bounded above. This leads to wide confidence bounds because the bounds have to account for the possibility of large changes to the mean that stem from unlikely draws of extreme values. When using importance sampling, we sample points according to some distribution $q$ and we can maintain an unbiased estimate of $p_y$ by computing a weighted average of the indicator function, $\mathbf{1}(y_i = y)$, where each observation is weighted by its inverse propensity, which implies importance weights $w_i = \frac{1/n}{q_i}$. The resulting random variable $z_i = w_i \mathbf{1}(y_i = y)$ has expected value equal to $p_y$, but can potentially take on values in the range $[0, \max_i \mathbf{1}(y_i = y) \frac{1/n}{q_i}]$. Because the distribution $q$ has probability that is inversely proportional to distance, this range has a natural interpretation in terms of the quality of the embedding space: the largest $z_i$ is the example from our target class that is furthest from our example embedding. If the embedding space does a poor job of clustering senses around the example embedding, then it is possible that the furthest point in embedding space—which will have tiny propensity because propensity decreases exponentially with distance—shares the same label as our target class, so our bounds have to account for this possibility.

There are two ways one could tighten these bounds: either make assumptions on the distribution of senses in embedding space that imply clustering around the example embedding, or modify the sampling strategy. We choose the latter approach: we can control the range of the importance weighted samples by enforcing a minimum value, $\alpha$, on our sampling distribution $q$ such that the resulting upper bound $\max_i \frac{1/n}{q_i} = \frac{\alpha^{-1}}{n}$. In practice this can be achieved by simply adding a small constant to each $q_i$ and renormalizing the distribution. Furthermore, we note that when the embedding space is relatively well clustered, the true distribution of $z$ will have far lower variance than the worst case implied by the bounds. We take advantage of this by computing our confidence intervals using Maurer & Pontil (2009)'s empirical Bernstein inequality which offers tighter bounds when the empirical variance is small.

**Lemma 2** (Empirical Bernstein). *Let* $z_i$ *be a sequence of i.i.d. bounded random variables on the range* $[0,m]$ *with expected value* $p_y$, *empirical mean* $\bar{z} = \frac{1}{n}\sum_i z_i$, *empirical variance* $V_n(Z)$. *For any* $\delta \in (0,1)$ *we have,*

$$\mathbb{P}\left[p_y \geq \bar{z} + \sqrt{\frac{m^2 2V_n(Z)\log(2/\delta)}{n}} + \frac{7m\log(2/\delta)}{3(n-1)}\right] \leq \delta$$

*Proof.* Let $z_i' = \frac{z_i}{m}$ such that it is bounded on $[0,1]$. Apply Theorem 4 of Maurer & Pontil (2009) to $z_i'$. The result follows by rearranging to express the theorem in terms of $z_i$. $\square$

The bound is symmetric so the lower bound can be obtained by subtracting the interval. Note that the width of the bound depends linearly on the width of the range. This implies a practical trade-off: making the $q$ distribution more uniform by increasing the probability of rare events leads to tighter bounds on the class frequency which rules out rare classes more quickly; but also costs more by selecting sub-optimal points more frequently.

**Unknown classes**  One may also want to allow for the possibility of labelling unknown classes that do not appear in the dictionary or lexical database. We use the following approach. If during sampling we discover a new class, it is treated in the same manner as the known classes, so we maintain estimates of its empirical frequency and associated bounds. This lets us optimize for classifier uncertainty over the new class and remove it from consideration if at any point the upper

bound on its frequency falls below our threshold (which may occur if the unknown class simply stems from an annotator misunderstanding).

For the stronger guarantee that with probability $1 - \delta$ we have found all classes above the threshold, $\gamma$, we need to collect at least $n \geq \frac{\log 1/\delta}{\gamma^2}$ uniform at random samples.

**Lemma 3.** *For any $\delta \in (0,1)$, consider an algorithm which terminates if $\sum_i \mathbf{1}(y_i = k) = 0$ after $n \geq \frac{\log 1/\delta}{\gamma^2}$ draws from an i.i.d. categorical random variable, $y$, with support $\{1, \ldots, K\}$ and parameters $p_1, \ldots, p_K$. Let the "bad" event, $\{B_j = 1\}$, be the event that the algorithm terminates in experiment $j$ with parameter $p_k > \gamma$. Then $\mathbb{P}[B_j = 1] \leq \delta$, where the probability is taken across all runs of the algorithm.*

*Proof sketch.* The result is a direct consequence of Hoeffding's inequality. $\square$

A lower bound of this form is unavoidable without additional knowledge, so in practice we suggest using a larger threshold parameter $\gamma'$ for computing $n$ if one is only worried about 'frequent' unknown classes. When needed, we support these unknown class guarantees by continuing to use an $\epsilon$-greedy sampling strategy until we have collected at least $n(\delta, \gamma)$ uniform at random samples without encountering an unknown class.

**Regret analysis**   Regret measures loss with respect to some baseline strategy, typically one that is endowed with oracle access to random variables that must be estimated in practice. Here we define our baseline strategy to be that of an active learning algorithm that knows in advance which of the classes fall below the threshold $\gamma$. This choice of adversary lets us focus our analysis on the affect of searching for thresholded classes.

Algorithm 1 employs the same strategy as the optimal agent during both the search and active learning phases, but may differ in the set of classes that it considers active: in particular, at the start of execution, it considers all classes active whereas the optimal strategy only regards all classes for which $p_{y_i} > \gamma$ as active. Because of this it will potentially remain in the exemplar guided search phase of execution for longer than the optimal strategy and hence the sub-optimality of Algorithm 1 will be a function of the number of samples it takes to rule out classes that fall below the threshold $\gamma$.

Let $\Delta = \min_i |p_i - \gamma|$ denote the smallest gap between $p_i$ and $\gamma$. Assume the classifier's performance on class $y$ can be described by some concave utility function, $U : \mathbb{R}^{n \times d} \times [K] \to \mathbb{R}$, that measures expected generalization as a function of the number of observations it receives. This amounts to assuming that standard generalization bounds hold for classifiers trained on samples derived from the active learning procedure.

**Theorem 4.** *Given some finite time horizon $n$ implied by the labelling budget, if the utility derived from additional labeled examples for each class $y$ can be described by some concave utility function, $U : \mathbb{R}^{n \times d} \times [K] \to \mathbb{R}$ and $\Delta = \min_i |p_{y_i} - \gamma| > 0$ then Algorithm 1 has regret at most, $\mathcal{R} \leq 1 + k(\gamma) \left[ \frac{2 \log(n)}{\Delta^2} + \frac{2 + \Delta^2}{n\Delta^2} \right]$*

The proof leverages that fact that the difference in performance is bounded by differences in the number of times Algorithm 1 selects the unthresholded classes. Full details are given in the appendix.

## 4   Experiments

Our experiments are designed to test whether automated search with embeddings could find examples of very rare classes and to assess the effect of different skew ratios on performance.

**Reddit word sense disambiguation dataset**   The standard practice for evaluating active learning methods is to take existing supervised learning datasets and counterfactually evaluate the performance that would have been achieved if the data had been labeled under the proposed active learning policy. While there do exist datasets for word sense disambiguation (e.g Yuan et al., 2016; Edmonds & Cotton, 2001; Mihalcea et al., 2004), they typically either have a large number of words with few examples per word or have too few examples to test the more extreme label skew that shows the benefits of guided learning approaches. To test the effect of a skew ratio of 1:200 with 50 examples

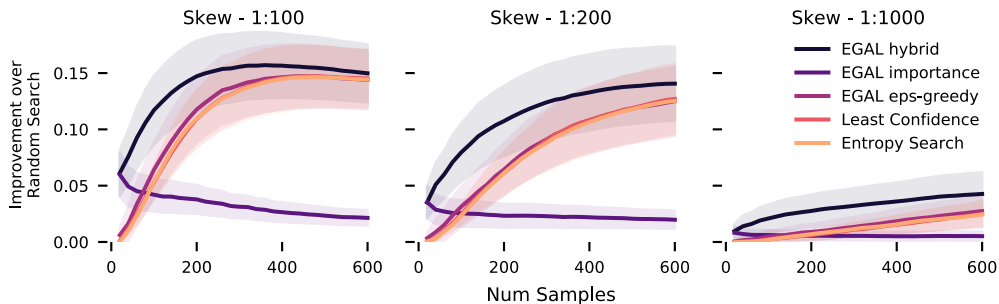

Figure 1: Average accuracy improvement over random search for all 21 words at different levels of skew. With lower levels of skew (*left*), EGAL tends to give big improvments over random search quickly as the examplars make it relatively easy to find examples of the rare classes. With larger amounts of skew (*left*), it takes longer on average before the uncertainty driven methods find examples of the rare class, so the performance difference with EGAL remains large for longer. Once skew becomes sufficiently large (*right*), EGAL still offers some benefit, but the absolute gains are smaller as the rare classes are suffiently rare that they are hard to find even with an exemplar.

of the rare class, one would need 10 000 examples of the common class; more extreme skew would require correspondingly larger datasets. The relatively small existing datasets thus limit the amount of label skew that is possible to observe, but as an artifact rather than a property of real text.

To address this, we collected a new dataset for evaluating active learning methods for word sense disambiguation. We took a large publicly-available corpus of Redditcomments (Baumgartner, 2015) and leveraged the fact that some words will exhibit a "[o]ne sense per discourse" (Gale et al., 1992) effect: discussions in different subreddits will typically use different senses of a word. Taking this assumption to the extreme, we label all applicable sentences in each subreddit with the same word sense. For example, we consider occurrences of the word "bass" to refer to the fish in the `r/fishing` subreddit, but to refer to low frequencies in the `r/guitar` subreddit. Obviously, this approach produces an imperfect labelling; e.g., it does not distinguish between nouns like "The bass in this song is amazing" and the same word's use as an adjective as in "I prefer playing bass guitar", and it assumes that people never discuss music in a fishing forum. Nevertheless, this approach allows us to evaluate more extreme skew in the label distribution than would otherwise have been possible. Overall, our goal was to obtain realistic statistical structure across word senses in a way that can leverage existing embeddings, not to maximize accuracy in labeling word senses.

We chose the words by listing the top 1000 most frequently used words in the top 1000 most commented subreddits, and manually looking for words whose sense clearly correlated with the particular subreddit. Table 1 lists the 21 words that we identified in this way and the associated subreddits that we used to determine labels. For each of the words, we used an example sentence from each target sense from Lexico as exemplar sentences.

**Setup**   All experiments used Scikit Learn (Pedregosa et al., 2011)'s multi-class logistic regression classifier with default regularization parameters on top of BERT (Devlin et al., 2018) embeddings of the target word. We used Huggingface's Transformer library (Wolf et al., 2019) to collect the `bert-large-cased` embeddings of the target word in the context of the sentence in which it was used. This embedding gives a 1024 dimensional feature vector for each example. We repeated every experiment with 100 different random seeds report the mean and $95\%$ confidence intervals[3] on test set accuracy. The test set had an equal number of samples from each class that exceeded the threshold.

**Results**   We compare three versions of the Exemplar-Guided Active Learning (EGAL) algorithm relative to uncertainty driven active learning methods (Least Confidence and Entropy Search) and random search. Figure 1 gives aggregate performance of the various approaches, aggregated across 100 random seeds and the 21 words. We found the hybrid of importance sampling for guided search

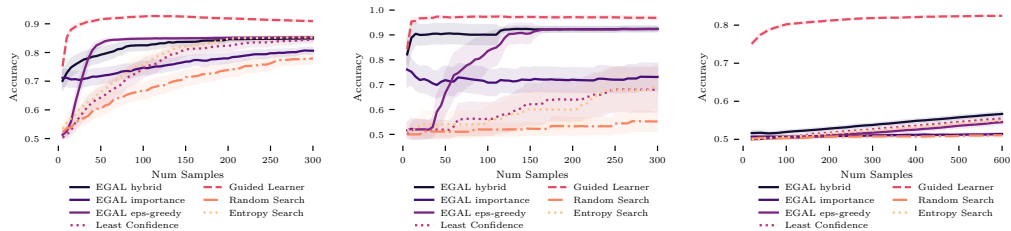

Figure 2: Accuracy vs number of samples for *bass* (left), *bank* (middle) and *fit* (right), having label skew of 1:60, 1:450 and 1:100 respectively. The word *bass* is a case where EGAL achieves significant gains with few samples; these gains are eventually evened out once the standard active learning methods gain sufficient samples. *Bank* has both a high quality exemplar and extreme skew, leading to large gains by using EGAL. *Fit* shows a failure case where EGAL's performance does not differ significantly from standard approaches.

and $\epsilon$-greedy active learning worked best across a variety of datasets. This EGAL hybrid approach outperformed all baselines for all levels of skew, with the largest relative gains at 1:200: with 100 examples labeled, EGAL had an increase in accuracy of 11% over random search and 5% over the active learning approaches.

In Figure 2 we examine performance on three individual words and include guided learning as an oracle upper bound on the performance improve that could be achieved by a perfect exemplar search routine. On average guided learning achieved over $80 - 90\%$ accuracy on a balanced test for both tasks using less than ten samples. By contrast, random search achieved $55\%$ and $80\%$ accuracy on *bass* and *bank* respectively, and did not perform better than random guessing on *fit*. This suggests that the key challenge for all of these datasets is collecting balanced examples. For the first two of these three datasets, having access to an exemplar sentence gave the EGAL algorithms a significant advantage over the standard approaches; this difference was most stark on the *bank* dataset, which exhibits far more extreme skew in the label distribution. On the *fit* dataset, EGAL did not significantly improve performance, but also did not hurt performance. These trends were typical (see the appendix for all words): on two thirds of the words we tried, EGAL achieved significant improvements in accuracy, while on the remaining third EGAL offered no significant improvements but also no cost as compared to standard approaches. As with guided learning, direct comparisons between the methods are not on equal footing: the exemplar classes give EGAL more information than the standard methods have access to. However, we regard this as the key experimental point of our paper. EGAL provides a simple approach to getting potentially large improvements in performance when the label distribution is skewed, without sacrificing performance in settings where it fails to provide a benefit.

## 5  Conclusions

We present the Exemplar Guided Active Learning algorithm that leverages the embedding spaces of large scale language models to drastically improve active learning algorithms on skewed data. We support the empirical results with theory that shows that the method is robust to mis-specified target classes and give practical guidance on its usage. Beyond word-sense disambiguation, we are now using EGAL to collect multi-word expression data, which shares the extreme skew property.

## Broader Impact

This paper presents a method for better directing an annotation budget towards rare classes, with particular application to problems in NLP. The result could be more money spent on annotation because such efforts are more worthwhile (increasing employment) or less money spent on annotation if "brute force" approaches become less necessary (reducing employment). We think the former is more likely overall, but both are possible. Better annotation could lead to better language models, with uncertain social impact: machine reading and writing technologies can help language learners and knowledge workers, increasing productivity, but can also fuel various negative trends including misinformation, bots impersonating humans on social networks, and plagiarism.

## Footnotes

[1]Example usages can be found in dictionaries or other lexical databases such as WordNet.

[2]For the probability of seeing at least one example to exceed $1 - \delta$, we need at least $n \geq \frac{\log 1/\delta}{\gamma^2}$ samples. See Lemma 3 for details.

[3]Computed using a Normal approximation to the mean performance.

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
