[Supplementary Material · 2020_NeurIPS__Exemplar_Guided_Active_Learning-2.pdf]

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

$\quad\quad\quad \gamma :$ the label-probability threshold
$\quad\quad\quad E$ the set of exemplars, $|E| = L$
$\quad\quad\quad B :$ a budget for maximum number of queries
$\quad\quad\quad b :$ batch size of queries sampled before the model is retrained

$\mathcal{A} \leftarrow \{1, \dots, L\}$ # Set of active classes
$\mathcal{C} \leftarrow \emptyset$ # Set of completed classes
$\mathcal{D}^{(l)} \leftarrow \emptyset$ # Set of labeled examples
**while** $|\mathcal{D}^{(l)}| < B$ **do**
$\quad\quad \mathcal{A}' = \mathcal{A}$ # $\mathcal{A}'$ is the target set of classes for the algorithm to find.
$\quad\quad$ **while** $\mathcal{A}' \neq \emptyset$ *and number of collected samples $< b$* **do**
$\quad\quad\quad$ Select random $i'$ from $\mathcal{A}'$ and set $\mathcal{A}' \leftarrow \mathcal{A}' \setminus \{i'\}$ and $X \leftarrow \emptyset$
$\quad\quad\quad$ **repeat**
$\quad\quad\quad\quad X \leftarrow X \cup \{x\}$ where $x$ is selected with exemplar $E_{i'}$ using either equation 1 or
$\quad\quad\quad\quad\quad \epsilon$-greedy sampling.

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

# A Appendix

## A.1 Analysis

The analysis of our regret bound uses the confidence bounds implied by Hoeffding's inequality which we state below for completeness. We focus on the $\epsilon-$greedy instantiation of the algorithm which gives estimates of $\hat{p}_y$ which are bounded between $0$ and $1$. The tighter confidence intervals used in Algorithm 1 improve constants but at the expense of clarity.

**Lemma 5** (Hoeffding's's inequality). *Let $X_1, ..., X_n$ be independent random variables bounded by the interval $[0, 1] : 0 \leq X_i \leq 1$ with mean $\mu$ and empirical mean $\bar{X}$. Then for $\epsilon > 0$, $\mathbb{P}\left[\bar{X} - \mu \geq \epsilon\right] \leq \exp(-2n\epsilon^2)$.*

Before proving Theorem 1, we give a lemma that bounds the probability that the active set of Algorithm 1 is non-empty after $T$ steps.

**Lemma 6.** *Let $T_y(t)$ denote the number of times we have observed an example from class $y$ after $t$ steps, and $\mathcal{A}_t = \{y : T_y(t) = 0, \hat{p}_y + \sigma_y > \gamma\}$ denote the set of "active" classes with upper confidence bound $\hat{p}_y + \sigma_y$ above the threshold. Let $\Delta = \min_i |p_i - \gamma|$ and $T = \left\lceil \frac{\log(1/\delta)}{2(1-c)^2\Delta^2} \right\rceil$ for some $c \in (0, 1)$ and $\delta \in (0, 1)$. Then for all $t > T$, $\mathbb{P}\left[\mathcal{A}_t \neq \emptyset\right] \leq \exp(-2tc^2\Delta^2)$*

*Proof.* By the choice of $T$, for all $c$ and $t \geq T$,

$$\Delta - \sqrt{\frac{\log(1/\delta)}{2t}} \geq c\Delta. \tag{4}$$

Now,

$$
\begin{aligned}
\mathbb{P}\left[\mathcal{A}_t \neq \emptyset\right] &= \mathbb{P}\Big[\exists y \quad s.t. \quad \{T_y(t) = 0\} \\
&\quad \cap \{\hat{p}_y(t) + \sqrt{\frac{\log(1/\delta)}{2t}} > \gamma\}\Big] \\
&\leq \mathbb{P}\left[\max_{y:\hat{p}_y<\gamma} \hat{p}_y(t) + \sqrt{\frac{\log(1/\delta)}{2t}} > \gamma\right] \\
&= \mathbb{P}\left[\hat{p}_{y_{\max}}(t) - p_{y_{\max}} > \Delta_{y_{\max}} - \sqrt{\frac{\log(1/\delta)}{2t}}\right] \\
&\leq \mathbb{P}\left[\hat{p}_{y_{\max}}(t) - p_{y_{\max}} > c\Delta\right] \\
&\leq \exp(-2tc^2\Delta^2)
\end{aligned}
$$

$\square$

The first inequality follows by dropping the $\{T_y(t) = 0\}$ event and noting that if $\{\hat{p}_y(t) + \sqrt{\frac{\log(1/\delta)}{2t}} > \gamma\}$ then the upper confidence bound of the largest $\hat{p}_y(t)$ estimate must also exceed the threshold. We denote this estimate $\hat{p}_{y_{\max}}(t)$ and its corresponding $\Delta_{y_{\max}}$ values and true means $p_{y_{\max}}$ analogously. The second inequality follows from Equation 4 the fact that $\Delta_{y_{\max}} \geq \Delta$. Finally, the third inequality applies Hoeffding's inequality (Lemma 5) with $\epsilon = c\Delta$.

## A.2 Proof of Theorem 4

*Proof.* Because the loss function is 0 for all classes that fall below the threshold, the difference in performance between the oracle and Algorithm 1 is only a function of the difference between the number of times each algorithm samples the unthresholded classes. Let $n$ denote a finite horizon, $T_y(t)$ denote the number of times that an example from class $y$ is selected by Algorithm 1 after $t$ iterations and define the event that the algorithm does not treat a class as thresholded after $n$ iterations as, $E_y = \{\min_{t \in [n]} \hat{p}_y + \sqrt{\frac{\log(1/\delta)}{2t}} \geq \gamma\}$.

By summing over the unthresholded classes, we can decompose regret as follows,

$$
\begin{aligned}
\mathcal{R} = & \sum_{y:p_y \geq \gamma} \mathbb{E}\left[\mathbf{1}(E_y^c)\left[U(T_y^*(n)) - U(T_y(n))\right]\right] + \\
& \qquad \mathbb{E}\left[\mathbf{1}(E_y)\left[U(T_y^*(n)) - U(T_y(n))\right]\right] \\
\leq & \sum_{y:p_y \geq \gamma} \left(\mathbb{P}\left[E_y^c\right]U(T_y^*(n)) + \right. \\
& \qquad \left. \mathbb{E}\left[U(T_y^*(n)) - U(T_y(n))\Big|E_y\right]\right)
\end{aligned}
$$

where the first term measures regret from falsely declaring the class as falling below the threshold, and the second accounts for the difference in the number of samples selected by the oracle and Algorithm 1. The inequality follows by upper-bounding the difference between the oracle and algorithm, $U(T_y^*(n)) - U(T_y(n))$, with $U(T_y^*(n))$.

To bound this regret term, we show the first term is rare and the latter results in bounded regret. First, consider the events for which the class is falsely declared as being below the threshold. We bound this by noting that for all $t \in [n]$,

$$
\begin{aligned}
& \mathbb{P}\left[\hat{p}_y + \sqrt{\frac{\log(1/\delta)}{2t}} < \gamma \Big| p_y > \gamma\right] \\
\leq & \mathbb{P}\left[\hat{p}_y + \sqrt{\frac{\log(1/\delta)}{2t}} < p_y \Big| p_y > \gamma\right] \leq \delta
\end{aligned}
$$

The first inequality uses the fact that $p_y > \gamma$ and the second applies the concentration bound.

In the complement of this event, the class is correctly maintained among the active classes so all we have to consider is how far the expected number of times that class is selected deviates from the number of times it would have been selected under the optimal policy. By assuming a concave utility function, we are assuming diminishing returns to the number of samples that you collect for each sample. This implies that we can bound $U(T_y^*(n)) - U(T_y(n)) \leq T_y^*(n) - T_y(n)$.

Let $t'$ denote the iteration in which the last class is removed from the active set, such that $\mathcal{A}_{t'-1} \neq \emptyset$ and $\mathcal{A}_{t'} = \emptyset$. This is the step at which regret is maximized since it is the point at which the two approaches have the largest differences in the number of samples for the unthresholded states. For any subsequent step the Algorithm 1 is unconstrained in its behavior (since it no longer has to sample) and can make up some of the difference in performance with the oracle because there are diminishing returns to performance from collecting more samples. Note that for any class $y$ with $p_y > \gamma$ that is correctly retained, we know that $\hat{p}_y - \sqrt{\frac{\log(1/\delta)}{2t'}} > \gamma$ and so it must have been selected at least $\gamma t'$ times. The optimal strategy would have selected the class at most $\frac{t'}{k(\gamma)}$ times, where $k(\gamma)$ is the number of classes that exceed the threshold, so the expected difference between $T_y(t')$ and $T_y^*(t')$ is at most $\left(\frac{1}{k(\gamma)} - \gamma\right)t' = f(\gamma)t'$ for some $f(\gamma) \in [0,1]$ that depends only on the choice of $\gamma$.

Because of this, bounding under-sampling regret reduces to bounding, $t'$, the number of uniform samples the algorithm draws before all class are declared inactive. To satisfy the conditions of Lemma 6, assume the algorithm draws at least $T > \frac{\log(1/\delta)}{2(1-c)^2\Delta^2}$ uniform samples, and thereafter $\mathbb{P}\left[\mathcal{A}_t \neq \emptyset\right] \leq 2\exp(-2tc^2\Delta^2)$.

$$\mathbb{E}\left[t'\right] \leq T + \sum_{t=T+1}^{n} \mathbb{P}\left[\mathcal{A}_{t-1} \neq \emptyset \cap \mathcal{A}_t = \emptyset\right]$$

$$\leq T + \sum_{t=T+1}^{n} \mathbb{P}\left[\mathcal{A}_{t-1} \neq \emptyset\right]$$

$$\leq T + \sum_{t=T+1}^{n} \exp(-2tc^2\Delta^2)$$

$$\leq T + \exp(-2c^2T\Delta^2)\frac{1}{1 - \exp(-2c^2\Delta^2)}$$

$$\leq T + \exp(-2c^2T\Delta^2)[\frac{1}{2c^2\Delta^2} + 1]$$

The second last inequality uses the fact that the sum is a geometric series (and takes $n \to \infty$), and the final inequality uses the identity $\frac{1}{1-\exp(-a)} \leq 1 + \frac{1}{a}$ for all $a$.

Putting this together, we get,

$$\mathcal{R} \leq \sum_{y:p_y \geq \gamma} \left( \mathbb{P}\left[E_y^c\right] U(T_y^*(n)) + \right.$$

$$\left. \mathbb{E}\left[U(T_y^*(n)) - U(T_y(n)) \Big| E_y\right]\right)$$

$$\leq k(\gamma) \left[\frac{n}{k(\gamma)}\delta + T + \exp(-2(T)c^2\Delta^2)[\frac{1}{2c^2\Delta^2} + 1]\right]$$

$$= 1 + k(\gamma)[\frac{2\log(n)}{\Delta^2} + \frac{2 + \Delta^2}{n\Delta^2}]$$

Where inequality uses the results above and the equality follows from substituting the expression for $T$, collecting like terms, setting $\delta = \frac{1}{n}$ and $c = \frac{1}{2}$, and collects like terms.

$\square$

## B  Experimental details

Table 1 shows all the words used for the experiments, as well as the target subreddits, exemplar classes and the number of examples from each of the classes. We subsampled the rare classes in order to achieve target skew ratios for the datasets.

We collected the words by extracting all usages of the target words in the target subreddits from the months of January 2014 to the end of May 2015. This required parsing 450GB of Baumgartner dataset. All the embeddings were collected using a single GPU and the active learning experiments were run on the CPU nodes of a compute cluster with $< 30$ nodes.

## C  Additional experiments

### C.1  Synthetic data for embedding quality

Because EGAL depends on embeddings to search the neighbourhood of a given exemplar, it is likely that performance depends on the quality of the embedding. We would expect better performance from embeddings that leads to better separated the classes because as classes become better separated, it become correspondingly more likely that a single exemplar will locate the target class. To evaluate this, we constructed a synthetic 'skew MNIST' dataset as follows: the training set of the MNIST dataset is subsampled such that the classes have empirical frequency ranging from $0.6\%$ of the

| Word | Sense 1 | n | Sense 2 | n | Sense 1 Exemplar | Sense 2 Exemplar |
|---|---|---|---|---|---|---|
| back | r/Fitness | 106764 | r/legaladvice | 18019 | He had huge shoulders and a broad back which tapered to an extraordinarily small waist. | Marcia is not interested in getting her job back, but wishes to warn others. |
| bank* | r/personalfinance | 22408 | r/canoeing, r/fishing, r/kayaking, r/rivers | 113 | That begs the question as to whether our money would be safer under the mattress or in a bank deposit account than invested in shares, unit trusts or pension schemes. | These figures for the most part do not include freshwater wetlands along the shores of lakes, the bank of rivers, in estuaries and along the marine coasts. |
| bass | r/guitar | 3237 | r/fishing | 1816 | The drums lightly tap, the second guitar plays the main melody, and the bass doubles the second guitar | They aren't as big as the Caribbean jewfish or the potato bass of the Indo-Pacific region, though |
| card | r/personalfinance | 130794 | r/buildapc | 62499 | However, if you use your card for a cash withdrawal you will be charged interest from day one. | Your computer audio card has great sound, so what really matters are your PC's speakers. |
| case | r/buildapc | 128945 | r/legaladvice | 22966 | It comes with a protective carrying case and software. | The cost of bringing the case to court meant the amount he owed had risen to £962.50. |
| club | r/soccer | 113743 | r/golf | 16831 | Although he has played some club matches, this will be his initial first-class game. | The key to good tempo is to keep the club speed the same during the backswing and the downswing. |
| drive | r/buildapc | 52061 | r/golf | 48854 | This means you can record to the hard drive for temporary storage or to DVDs for archiving. | If we're out in the car, lost in an area we've never visited before, he would rather we drive round aimlessly for hours than ask directions. |
| fit | r/Fitness | 75158 | r/malefashionadvice | 16685 | The only way to get fit is to make exercise a regularly scheduled part of every week, even every day. | The trousers were a little long in the leg but other than that the clothes fit fine. |
| goals | r/soccer | 87831 | r/politics r/economics | 2486 | For the record, the Brazilian Ronaldo scored two goals in that World Cup final win two years ago. | As Africa attempts to achieve ambitious millennium development goals, many critical challenges confront healthcare systems. |
| hard | r/buildapc | 63090 | r/gaming | 34499 | That brings me to my next point: never ever attempt to write to a hard drive that is corrupt. | It's always hard to predict exactly what the challenges will be. |
| hero | r/DotA2 | 198515 | r/worldnews | 5816 | Axe is a more useful hero than the Night Stalker | As a nation, we ought to be thankful for the courage of this unsung hero who sacrificed much to protect society. |
| house | r/personalfinance | 46615 | r/gameofthrones | 11038 | Inside, the house is on three storeys, with the ground floor including a drawing room, study and dining room. | After Robert's Rebellion, House Baratheon split into three branches: Lord Robert Baratheon was crowned king and took residence at King's Landing |
| interest | r/personalfinance | 64394 | r/music | 1658 | The bank will not lend money, and interest payments and receipts are forbidden. | The group gig together about four times a week and have attracted considerable interest from record companies. |
| jobs | r/politics r/economics | 42540 | r/apple | 3642 | In Kabul, they usually have low-paying, menial jobs such as janitorial work. | Steve Jobs demonstrating the iPhone 4 to Russian President Dmitry Medvedev |
| magic | r/skyrim | 30314 | r/nba | 6655 | Commonly, sorcerers might carry a magic implement to store power in, so the recitation of a whole spell wouldn't be necessary. | Earvin Magic Johnson dominated the court as one of the world's best basketball players for more than a decade. |
| manual | r/cars | 2752 | r/buildapc | 563 | Power is handled by a five-speed manual gearbox that is beefed up, along with the clutch - make that a very heavy clutch. | What is going through my head, is that this guy is reading the instructions directly from the manual, which I can now recite by rote. |
| market | r/investing | 23563 | r/nutrition | 150 | It's very well established that the U.S. stock market often leads foreign markets. | I have seen dandelion leaves on sale in a French market and they make a tasty addition to salads - again they have to be young and tender. |
| memory | r/buildapc | 21103 | r/cogsci r/psychology | 433 | Thanks to virtual memory technology, software can use more memory than is physically present. | Their memory for both items and the associated remember or forget cues was then tested with recall and recognition. |
| ride | r/drums | 36240 | r/bicycling | 5638 | I learned to play on a kit with a hi-hat, a crash cymbal, and a ride cymbal. | It is, for example, a great deal easier to demonstrate how to ride a bicycle than to verbalize it. |
| stick | r/drums | 36486 | r/hockey | 2865 | Oskar, disgusted that the singing children are so undisciplined, pulls out his stick and begins to drum. | Though the majority of players use a one-piece stick, the curve of the blade still often requires work. |
| tank | r/electroniccigarette | 64978 | r/WorldofTanks | 47572 | There are temperature controlled fermenters and a storage tank, and good new oak barrels for maturation. | First, the front warhead destroys the reactive armour and then the rear warhead has free passage to penetrate the main body armour of the tank. |
| trade | r/investing | 18399 | r/pokemon | 5673 | This values the company, whose shares trade at 247p, at 16 times prospective profits. | Do you think she wants to trade with anyone |
| video | r/Music | 51497 | r/buildapc | 23217 | Besides, how can you resist a band that makes a video in which they rock their guts out while naked and flat on their backs? | You do need to make certain that your system is equipped with a good-quality video card and a sound card. |

Table 1: Target words with associated subreddits and exemplar sentences. *For the "bank" word, we manually removed all comments of the form, "this won't break the bank", that we would otherwise have classified as referring to river banks.

Figure 3: Accuracy for for the methods evaluated with different quality embeddings simulated by training a convolution network for 0, 1 and 10 epochs respectively. From left to right we see the difference in performance with improving quality of embedding. The left plot measures performance of logistic regression on the features from the final hidden layer of a randomly initialized convolutional network. The middle is performance after the convolutional network has been trained for a single epoch, and right is after 10 epochs. EGAL offers clear performance improvements with high quality embeddings and no degradation in performance relative to the standard approaches with poor quality embeddings.

Figure 4: Accuracy as we vary the number of rare examples for the *bass*. The left plot has 50 rare examples, the middle plot has 100 and the right plot has 400. As the dataset becomes more balanced, the advantage of having access to an exemplar embedding is diminished.

data (101 training examples) to $37.4\%$ (6501 training examples) for the most frequent class. The distribution was chosen to be as skew as possible given that the most frequent MNIST digit has 6501 training examples. The test set was kept as the original MNIST test set which is roughly balanced, so as before, the task is to transfer to a setting with balanced labels.

We evaluated the effect of different quality embeddings on performance, by using the final hidden layer of a convolution network trained for a different number of epochs on the full MNIST dataset. With zero epochs, this just corresponds to a random projection via a convolution network—so we would expect the classes to be poorly separated—but with more epochs the classes become more separable as the weights of the convolutional network are optimized to enable classification via the final layer's linear classifier. We evaluated the performance of a multi-class logistic regression classifier on embeddings from 0, 1 and 10 epochs of training.

Figure 3 shows the results. With a poor quality embedding that fails to separate classes, EGAL performed the same as the least confidence algorithm. As the quality of the embedding improved, we see the advantage of early samples from all the classes: the guided search procedure lead to class separation in very few samples, which can then be refined via the least confidence algorithm.

## C.2 Performance under more moderate skew

Figure 4 shows the performance of the various approaches as the label distribution skew becomes less dramatic. The difference between the various approaches is far less pronounced in this more benign case where there was no real advantage to having access to an exemplar embedding because random sampling will quickly find the more rare examples.

## C.3 Imbalance

We can gain some insight into what is driving the relative performance of the approaches by examining statistics that summarize the class imbalance over the duration of algorithms's executions. We measure imbalance using the following $R^2$-style statistic,

$$\text{Imbalance}(p) := 1 - \frac{\text{KL}(p, q_{\text{uniform}})}{\text{KL}(q_{\text{empirical}}, q_{\text{uniform}})}$$

where $q_{\text{empirical}}$ denotes the empirical distribution of target labels, $q_{\text{uniform}}$ is the uniform distribution over target labels and $\text{KL}(p, q) = -\sum_x p(x) \log \frac{p(x)}{q(x)}$ is the Kullback–Leibler divergence. The score is well-defined as long as the empirical distribution of senses is not uniform, and is calibrated such that any algorithm that returns a dataset that is as imbalanced as sampling uniformly at random attains a score of zero, while algorithms that perfectly balance the data attain a score of 1.

The plots in section D.2 show this imbalance score for each of the words. There are a number of observations worth noting:

1. The algorithms with the highest accuracy in figure 2 and section D.1, also produced a more balanced distribution of senses. Guided learning was naturally the most balanced of the approaches as it has oracle access to the true labels and it explicitly optimizes for balance.

2. The importance-weighted EGAL approach over-explores leading to more imbalanced sampling that behaves like uniform sampling. This explains its relatively poor performance.

3. The standard approaches are more imbalanced because it took a large number of samples for them to see examples of the rare class; once such examples were found, they tended to produce more balanced samples as they selected samples in regions of high uncertainty.

## C.4 Class coverage

The fact that the standard approaches needed a large number of samples to observe rare classes can also be seen in the plots in section D.3 which show the proportion of the unthresholded classes ($p_y \geq \gamma$) with at least one example after a given number of samples have been drawn by the respective algorithms. Again, we see that the standard approaches have to draw a far more samples before observing the rare classes because they don't have an exemplar embedding to guide search. These plots also give some insight into the failure cases. For example, if we compare the word `manual` to the word `goal` in figure 6 and figure 16 (which show accuracy and coverage respectively), we see that the exemplar for the word `goal` resulted in rare classes being found more quickly than in `manual`. This difference is reflected in a large early improvement in accuracy for EGAL methods over the standard approaches. This suggests that—as one would expect—EGAL offers significant gains when the exemplar leads to fast discovery of rare classes.

# D   All individual words

## D.1   Accuracy

Figure 5: Average accuracy for each of the individual words with a ratio of frequent to rare class of 1:100. The rare class is randomly sub-sampled to achieved the desired skew level; each experiments is repeated 100 times with a different sub-sample drawn for each random seed.

Figure 6: Average accuracy for each of the individual words with a ratio of frequent to rare class of 1:200.

Figure 7: Average accuracy for each of the individual words with a ratio of frequent to rare class of 1:1000.

## D.2 Imbalance

Figure 8: Average imbalance for each of the individual words with a ratio of frequent to rare class of 1:100.

Figure 9: Average imbalance for each of the individual words with a ratio of frequent to rare class of 1:200.

Figure 10: Average imbalance for each of the individual words with a ratio of frequent to rare class of 1:1000.

## D.3 Coverage

Figure 11: Average class coverage for each of the individual words with a ratio of frequent to rare class of 1:100.

Figure 12: Average class coverage for each of the individual words with a ratio of frequent to rare class of 1:200.

Figure 13: Average class coverage for each of the individual words with a ratio of frequent to rare class of 1:1000.