[Reviews · NeurIPS 2020]

Review 1

Summary and Contributions: The paper proposes a framework that identifies rare and common classes from the dataset using word embedding techniques and incorporates active learning techniques to label instances belonging to rare classes. The proposed method is an interesting way to identify exemplars of word senses in different contexts by employing the distance between contextual word representations of the word vectors. Once the algorithm determines exemplars of common classes, it leverages a standard active learning based framework to learn low entropy instances.

Strengths: The authors have proposed an intuitive method to find rare and common class instances by using contextualized word representations. The method is simple but effective for finding example usage of words for each sense. The authors provide some theoretical ground for choosing a tight confidence bound for finding the stopping rule, labelling unknown classes and regret bound.

Weaknesses: Although the approach is relevant and has some applications, the assumption that the test set contains common classes only seems far-fetched. Ignoring the tail label clases is a worrisome assumption. Even though one may assume that the classifier loss can be sufficiently improved by finding common classes, however, exemplars find significant use for the tail classes. More analysis on using contextualised representation to find similar word senses would be interesting. Currently, it is assumed that the BERT based architectures produce sufficiently good contextual representations. For example, the authors could run a sanity check experiment on how effective is the BERT embedding for determining the neighborhood examples. The experimental section must be broadened by including more datasets and results. Currently, experiments are performed on a single reddit dataset.

Correctness: Yes

Clarity: While the paper is well-written, the specific challenges that the authors claim to overcome need to be explained and illustrated better.

Relation to Prior Work: Yes, the prior work section is well discussed.

Reproducibility: Yes

Additional Feedback:


Review 2

Summary and Contributions: This paper presents an algorithm for efficiently labeling a data set for the purpose of word sense disambiguation. The algorithm consists of two steps first a search phase aimed at estimating the frequency of labels in the corpus. This phase is used to define the actual label set and prunes potential labels that are not above a given threshold. More specifically the algorithm starts with a list of potential word senses and a canonical example for each. They define a distribution that allows them to sample from the neighborhood of each of these examples while estimating the frequency of each potential word sense. They define efficient stopping rules for when to stop considering candidate labels which have probabilistic guarantees for correctness. At the end of this phase they have a fixed label set with samples from each category. The second phase is a more standard active learning algorithm aimed at requesting labels in order to improve classifier accuracy as efficiently as possible. The features used for both phases are embeddings taken from large language models. They develop a new word sense disambiguation data set from top reddit threads which is used to test their algorithm.

Strengths: The authors address an interesting problem of how to efficiently label a corpus for the purpose of word sense disambiguation. The authors provide a thorough analysis of their algorithm providing probabilistic guarantees and consider how to optimize key hyper-parameters. The empirical evaluation is convincing that their algorithm is achieving superior performance compared to simple but standard active learning baselines. I really liked the comparisons with how an ideal guided algorithm would perform. It shows that their algorithm while far from the ideal algorithm is achieving relatively strong performance.

Weaknesses: The empirical evaluation is somewhat limited given the fact that they only show results on a single data set and the baselines compared to are rather simple. This is understandable given the fact that the available datasets are limited for studying this problem. I wonder if there is another data set with rare labels and high amounts of skew that you can test this algorithm on. Even something artificial like creating a highly skewed version of cifar-100 or some other data set with many labels could be interesting. Also the first phase of the algorithm seems to really depend on the assumption that the differences in the embedding space are strongly associated with word sense. The authors referred to prior work that showed these embeddings are generally useful for downstream tasks. I think for the most part this is true, but the authors made very specific claims about bounds on frequency estimates and stopping conditions and it is not clear to me how well these claims would hold up to even to minor violations of this assumptions. And I'm not sure this would be reflected in the experiments provided given that they take place after the label pruning has already been performed. Essentially how do you know your label pruning in phase 1 is actually doing a good job? Could this easily be verified by showing a plot of well the frequency estimates actually match the ground truth in the data set?

Correctness: Yes, the claims and empirical methodology are correct.

Clarity: For the most part the paper is clearly written. I think the ordering of section 3 could be improved by discussing the stopping conditions immediately after the discussion on guided search and before the discussion of the active learning phase. Essentially I think what exactly what phase 1 is doing could be more clear. This is probably a minor point.

Relation to Prior Work: The authors address prior work in both word sense disambiguation, guided learning and active learning and what distinguishes their work.

Reproducibility: Yes

Additional Feedback:


Review 3

Summary and Contributions: The authors present a strategy to search for examples of rare classes can be found by 1) taking them from existing lists (e.g. dictionaries, for the case of word sense disambiguation, which the paper focuses on), 2) further expanding the set by searching for similar senses in embedding space (of e.g. BERT). Exemplars are identified using sampling, which enables estimating empirical frequencies of senses, determine whether they are rare enough to be ignored, and provide bounds on the quality of the solution upon termination. This strategy is used to initialize a more standard active learning setup, in a manner not too dissimilar from guided learning. UPDATE: I acknowledge reading the other reviews and the rebuttal. I am still convinced that this is a good paper, well worth accepting. For this reason, I kept my score unchanged.

Strengths: - Very well written and argumented. - The solution strategy is intuitive and appealing. - Welcome theoretical bounds. - Extensive experiments. In general I really enjoyed reading this paper.

Weaknesses: - It is unfortunate that for extreme skew this strategy does not provide as good of a performance increase. Not a big deal for me, the advantages are clear anyway. - If the rare classes have a high cost, the proposed strategy does not apply anymore. This should be stated clearly. Perhaps a version that estimates *low-cost* rather than *rare* classes should be developed for more general usage, but then probably the way in which the embeddings are built should be adjusted to take cost into consideration somehow.

Correctness: I didn't find any mistakes in either the method, the theory (although I did not check all steps), or the experiments. line 121: p_y and gamma be swapped caption of Fig 1: "so the rare classes so the performance difference"

Clarity: The text is very well written. All ideas are clearly exposed.

Relation to Prior Work: To my knowledge, all relevant prior work has been discussed.

Reproducibility: Yes

Additional Feedback:


Review 4

Summary and Contributions: This paper proposes new active learning methods for a type of word sense disambiguation problem. In the problem, we assume that an example for each sense is available at the beginning. Furthermore, the class/sense which occurs less than a threshold will be ignored in the test set, and the rest of classes/senses are equally important regardless of their frequency in the corpus. In order to estimate the class/sense frequency while collecting samples, the paper proposes three strategies: importance sampling, eps-greedy, and hybrid. The proposed methods have some nice theoretical guarantees (including regret bound) and have strong empirical performances.

Strengths: 1. The paper provides theoretical analysis on a practical problem (I am not a theoretical researcher, so I do not know how novel the theorem proposed by this paper is. I assume the analysis is correct and the contribution is substantial). 2. This paper describes complex theorem with a clear description and explains the intuition behind each design choice well. 3. The paper proposes a smart way to evaluate their methods. 4. The experimental results look strong (including the additional results in the supplementary materials).

Weaknesses: 1. It might be uncommon to have an application that needs to totally ignore classes with probability slightly lower than a threshold but emphasize the classes with probability slightly higher than the threshold. It makes some sense when we collect a word sense disambiguation dataset. I guess that is why the method is designed mostly for the word sense disambiguation problem, but an active learning method designed specifically for a NLP problem might have limited impact. (In the conclusion, it mentions that this method could also be applied to multi-word expression. I think this alleviates the concern a little.) In the broader impact section, I suggest the authors discuss some potential applications to further address this concern. 2. Although most of the parts in the paper are clear, the algorithm details are not clear to me (will elaborate in my clarity comment) and it is unclear why authors design the method as follow: "Once we have found an example of y, we stop searching for more examples and instead let further exploration be driven by classifier uncertainty." (please see the major question in my additional comments).

Correctness: I did not find obvious errors in the main paper, but I did not check the proof in the supplementary materials.

Clarity: 1. Each component is well written, but how every component is combined is not very clear. For example, how exactly the hybrid method combines eps-greedy and importance sampling (maybe I miss it somewhere in the paper)? The algorithm 1 is unclear. Does algorithm 1 describe the eps-greedy, importance sampling, or hybrid method? Where is i' used? What does "x is selected using equation 2" mean (I only saw S_ENT and S_LC in (2))? Where does it perform guide search? What does "Update stopping rule statistics" mean? Where is the stop rule applied in algorithm 1? Which lines in the algorithm 1 achieve "Once we have found an example of y, we stop searching for more examples and instead let further exploration be driven by classifier uncertainty."? 2. There seem to be some typos in supplementary materials. For example, proof of theorem 1 should be theorem 4. Figure C.1 seems to mean Figure 3. These typos confuse me a little.

Relation to Prior Work: 1. Related work could include active search research such as [1] 2. The limitation description of previous work could be improved. For example, when talking about Zhu & Hovy (2007), it says "did not address the problem of finding initial points under extremely skewed distributions". I got confused a little because this submitted work also assumes that an example is available at the beginning to guide the search. [1] Jiang, Shali, Roman Garnett, and Benjamin Moseley. "Cost effective active search." Advances in Neural Information Processing Systems. 2019.

Reproducibility: Yes

Additional Feedback: Major questions 1. Why is the sampling strategy switched to uncertainty sampling once an example is collected? Is it because after the classifier seeing one example in a rare class, it could start to give high uncertainty to the rare class? If that is the case, I do not understand why we cannot use the initial example (in WordNet?), which we assume to be available at the beginning, to train the classifier and directly use uncertainty sampling at the beginning. Minor questions: 1. Did you try to use cosine distance rather than L2 distance in the guided search? It might improve the performance a little? 2. Did you try to use the average non-contextized word embedding (e.g., word2vec) in the sentence containing the target word as the sense embedding x_i? Does BERT embedding for the target word outperform this more traditional baseline? 3. I think the explanation of how lambda_y is computed is unclear. Why solving (1) won't make lamda approach 0 (i.e., only sample the example with the closest embedding). For example, if we have 3 examples, 1 / (1*1+0+0) seems to be the minimum. Minor suggestions: 1. Mentioning word sense disambiguation in the title 2. Contextualized word embedding is a tool you used and your experiment actual does not show how effective using BERT is. I think citing them on introduction and/or experiment section is enough. You can remove them from the related work section in order to have more room to address the above two minor concerns. 3. The curve color and line style could be improved in the figure. It took me several seconds to find the curve corresponds to the random search in Fig. 2, and the colors in different curves could be more distinct. 4. I think NeurIPS typically use [number] style of citation. For the minor points, you do not need to address them in the rebuttal if you need more space to address other points (from other reviewers) because it probably won't change my overall evaluation of the paper. I think if my major question is answered properly and other reviewers do not point out some serious error, I will remain my acceptance vote and let the authors revise other minor flaws I mentioned in the future version of the paper. After seeing the rebuttal: Thanks for the rebuttal. My major question is properly addressed, so I remain my acceptance vote.

[Author Response · NeurIPS 2020]

Thank you to all the reviewers for their detailed reviews. We address specific concerns below.

**Reviewer 1**   *[test set contains common classes]* Thanks for pointing this out - we should have made this clearer: we are not claiming that an in-production test set would only contain common classes, but rather that the loss defined in line 118 gives zero weight to rare classes, which is mathematically equivalent to not having them in the test set. This amounts to saying, "my classifier should be equally good on all classes, except the extremely rare ones which we deem to matter at all". So if a word has a niche sense in some small community we do not penalize the classifier for not correctly classifying that sense; for example, if we know that an NLP system is not designed for technical conversations between mathematicians, we might not mind if our word sense system fails to recognize "group" as an algebraic structure so we give it zero loss if it fails on such a class in production.

*[how effective is the BERT embedding]* We agree that a thorough analysis of the word sense distribution in contextualized embeddings would be interesting, but it is beyond the scope of this project. That said, we can make some qualitative comments: at the start of this project we evaluated the embeddings by hand-labelling a couple of words and found BERT does a reasonably good job of separating classes (we explicitly leverage this observation by assuming we have a distance metric). Additionally, because we use a linear classifier on pre-trained BERT embeddings, one can also get some indication of how well separated the classes are by checking the accuracy for the oracle guided learning approach.

*[experimental section must be broadened]* See the response to Reviewer 2 below.

**Reviewer 2**   *[results from a single dataset]* As you point out, dataset availability is a challenge. We did perform an experiment along the lines of what you suggest with the Skew MNIST synthetic dataset in appendix C.1. We would be happy to include a similar experiment on CIFAR-100 if you think it would be valuable (note this in your final review if this is the case).

That said, we should note that this paper does more experiments than is typical in active learning: while the evaluated words share a data generating process, each word amounts to a different active learning problem. Typical active learning papers evaluate 10-15 active learning problems, whereas we have 21 words and the Skew MNIST dataset.

*[How does the method hold up to violations on assumptions regarding embedding quality?]* Good question - this was the motivation behind the the experiment in Appendix C1 which degrades the quality of the embedding on the Skew MNIST dataset. In short - we found that EGAL's performance degrades to that of the standard approaches as the embedding quality degrades (see paper for details).

**Reviewer 3**   Thank you for picking up those typos - we will correct them in the final draft. Regarding costs - that's a good point, we've assumed the cost of obtaining rare labels is driven by their rarity, but that the costs of labelling an individual example is uniform. We will make this clear in the camera ready.

**Reviewer 4**   *[Why is the sampling strategy switched to uncertainty sampling?]* Because the search phase searches the neighbourhood of the exemplar, once an example has been found, any additional examples from the target class will typically be very close in embedding space and hence provide relatively little marginal value. We treat the exemplars as out of distribution examples—example usage of a word sense from WordNet will typically differ from "in the wild" text in a Reddit corpus—in order to ensure that the final classifier is not biased by any covariate shift between example usages and actual usage[1]. Of course we need some similarity between the example usage and the "in the wild" usage for the exemplars to be projected into similar parts of embedding space, but this approach allows for differences between the distribution of the exemplars and that of the actual usage without introducing any bias into the final classifier.

*[Cosine distance]* Good suggestion, thanks!

*[Average non-contextized word embedding]* We only experimented with BERT embeddings. Performance clearly depends on the quality of the embedding space, so this is an important practical consideration; WordNet embeddings would also be far cheaper to compute. We will experiment with this.

We will incorporate your minor suggestions and include a more complete description of how $\lambda_y$ is computed in the text. Thank you for an extremely thorough review.

## Footnotes

[1]Note that in skew label distributions of the type we study, the classifier will typically see very few examples of the rare class even with the EGAL active learning strategy, so individual training examples can have a relatively large influence over the final decision boundary.


[Meta-Review · NeurIPS 2020]

The paper proposed to select unlabelled training examples based-on the embedding distance between the given exemplar and the query data. A pretrained BERT model is used to compute the embedding for the training examples. The problem formulation of selecting balanced labels in a highly skewed training set and the complexity bound is appreciated by all the reviewers. The general consensus is that the paper adds an interesting contribution to active learning methods applied to word sense disambiguation. The current version of the paper would be greatly strengthened by including more datasets. Also, the clarity of the paper could be improved by highlighting the proposed active learning method in an algorithm box.